# Finite-Time Neural Network Fault-Tolerant Control for Robotic Manipulators under Multiple Constraints

**Zhao Zhang, Lingxi Peng, Jianing Zhang and Xiaowei Wang ***

School of Mechanical and Electrical Engineering, Guangzhou University, Guangzhou 510006, China;
2111907017@e.gzhu.edu.cn (Z.Z.); penglx@gzhu.edu.cn (L.P.); zjn325@gzhu.edu.cn (J.Z.)
* Correspondence: meewxw_ee@gzhu.edu.cn

**Abstract:** In this study, a backstepping-based fault-tolerant controller for a robotic manipulator system with input and output constraints was developed. First, a barrier Lyapunov function was adopted to ensure that the system output satisfied time-varying constraints. Subsequently, the actuator input saturation and asymmetric dead-zone characteristics were also considered, and the actuator characteristics were described using a continuous function. The impacts of actuator failures and unknown dynamical parameters of the system were eliminated by employing Gaussian radial basis function neural networks. The external disturbances were compensated for, using a disturbance observer. Meanwhile, a finite-time dynamic surface technique was adopted to accelerate the convergence of the system errors. Finally, simulation of a 2-degrees-of-freedom robotic manipulator system showed the effectiveness of the proposed controller.

**Keywords:** actuator faults; input saturation; dead zone; output constraints; finite time

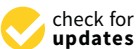

## 1. Introduction

Practical applications, such as robotic, electrical, and hydraulic systems, are frequently subject to input and output constraints, owing to their intrinsic characteristics. For instance, saturation, dead zone, hysteresis, and other input constraints have a direct impact on the response of the actuator [1–6], which affects system output performance. Meanwhile, the output or state constraints must be considered during the system controller design process to satisfy the system output performance criteria. However, these constraints do not exist individually. If these multi-constraint problems are disregarded during the design process, the system performance may deteriorate or fail. Thus, it is necessary to eliminate the influence of these nonlinear characteristics.

In recent years, numerous strategies to ensure the stability and performance of a system with dead-zone characteristics have been proposed [1,7–18]. For instance, to eliminate the influence of the dead-zone characteristic, in [7,8] an effective control was achieved by constructing a dead-zone inverse function, and the dead-zone characteristic was described by a linear function in [9,10]. An optimization algorithm was adopted in [19], where the authors treated the input dead-zone characteristic as a bounded function and adopted an adaptive approach to overcome the nonlinear characteristics. An adaptive fuzzy controller was utilized to compensate for the dead zone in [20], both of which avoided the problem of constructing the dead-zone inverse. With the development of neural networks (NNs) technology, in [21], for a flexible robotic arm system, the NNs approach was conquered based on the effect of dead-zone characteristics. In addition, owing to the performance requirements of the system, its output is subject to certain constraints [22–25]; the barrier Lyapunov function (BLF) is a common method for dealing with output or state constraint problems. In [26], a logarithmic BLF was applied to the control-law design of a robot. The control problem of a robotic arm system with time-varying output constraints was studied in [27], and an effective controller was presented for the model of a marine vessel under asymmetric constraints in [28], both of which expanded the application scenarios of the

BLF. However, only a single constraint was considered in the aforementioned works, and the constraints appeared in multiple forms in practical application scenarios. Therefore, the control law design under multiple constraints is worth investigating.

Furthermore, as the working time and environment change, the actuators in the system are prone to failure, which has an adverse impact on the system performance. Numerous successful outcomes have recently been demonstrated in terms of addressing the control challenges caused by actuator faults [29–40]. For instance, the effects of actuator failures and disturbances were handled using an adaptive-based strategy in [31] and eliminated using the NNs technique in [33]. Based on the backstepping strategy, the switched system overcame the effects of faults and achieved a fast convergence of errors in [34]. In [35,36], reasonable control laws were devised for the flexible robotic and spacecraft systems, respectively, and the system remained stable in the event of actuator failure without violating the constraints. In [37], an effective controller was designed to eliminate the effect of actuator failure by combining fuzzy and backstepping controls. However, in the aforementioned types of fault-tolerant control, the control problem under multiple constraints is rarely considered, and the combination of multiple constraints and actuator faults makes the controller design more challenging, thereby inspiring our research.

Inspired by the previous work, in this paper, based on the NNs and dynamic surface control (DSC) approach, we develop a suitable controller for a robotic system with multiple constraints and actuator failures. Compared with the existing work, the main contributions are as follows:

1. Compared to the results in [1,10,31,33,36], this study considered actuators with multiple constraints. The hyperbolic tangent function and asymmetric dead-zone function were introduced to describe the input characteristics of the system. The entire design process was based on the backstepping scheme in which the DSC and Nussbaum functions are utilized to optimize the design process.
2. In contrast to [11,33], time-varying output constraints were considered to ensure that the system still met the performance requirements, even if actuator failure occurred. The NNs approach was utilized to fit the unknown parameters and faults of the robots.
3. Based on [11], a finite-time filter was applied to optimize the design process and achieve fast convergence of the system error.

## 2. Problem Formulation

In this paper, we will study the problem of fault-tolerant control of a manipulator system with multiple constraints. The constraints considered include input saturation, dead zone, and time-varying output constraints.

Considering the uncertainty of the system structure and the possible actuator failures during operation, we employ radial basis function neural networks to compensate for the unknown continuous functions:

$$
\begin{aligned}
&f_i(Z) : \mathbb{R}^q \to \mathbb{R} \\
&f_i(Z) = W_i^T S_i(Z), \quad i = 1, 2, \ldots, n
\end{aligned}
\tag{1}
$$

where $W = [W_1, W_2, \ldots, W_l]^T \in \mathbb{R}^l$ is the weight vector, and $l$ is the number of nodes in the hidden layer of the NNs. Theoretically, when $l$ is chosen to be large enough, the output $W_i^T S_i(Z)$ of NNs can achieve an exact approximation for any continuous function. $Z = [Z_1, Z_2, \ldots, Z_q]^T \in \Omega_N \subset \mathbb{R}^q$ is the NNs input vector, and $S_i = [s_1, s_2, \ldots, s_l]^T \in \mathbb{R}^l$ is the output value of the activation function. The approximation process of the NNs is described as

$$
\begin{aligned}
&f_i(Z) = W_i^{*T} S_i(Z) + \varepsilon_i(Z) \quad \forall Z \in \Omega_N \subset \mathbb{R}^q \\
&i = 1, 2, \ldots, n
\end{aligned}
\tag{2}
$$

where $W^* = [W_1^*, W_2^*, \dots, W_l^*]^T$ is the vector of ideal weights, and $\varepsilon_i = [\varepsilon_1, \varepsilon_2, \dots, \varepsilon_l]^T$ is the approximation error that satisfies $|\varepsilon_i(Z)| \leq \bar{\varepsilon}_i$, where $\bar{\varepsilon}_i > 0$ is an unknown bound. The Gaussian-type activation function is

$$s_k(Z) = \exp\left[\frac{-(Z-\mu_k)^T(Z-\mu_k)}{\eta_k^2}\right], k = 1, 2, \dots, l \tag{3}$$

where $\mu_k = [\mu_{k1}, \mu_{k2}, \dots, \mu_{kq}]^T$ and $\eta_k$ denote the centers and widths of the Gaussian function, respectively. The structure used in this paper is a three-layer network, and it is shown in Figure 1.

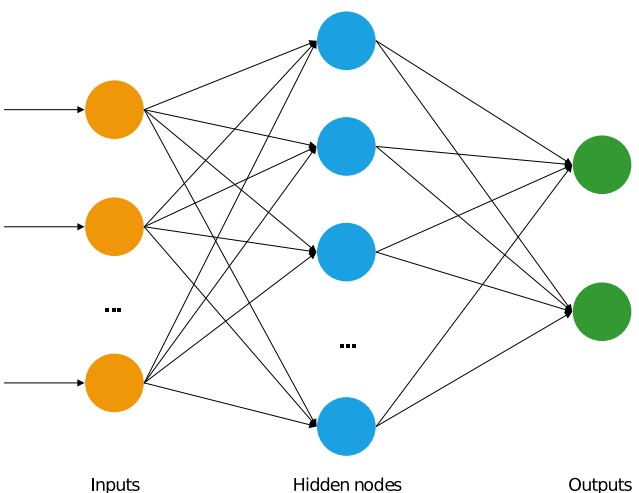

**Figure 1.** Schematic of neural network structure.

**Lemma 1** ([11]). *The following inequality holds for any pair of vectors $a, b \in R^n$.*

$$a^T b \leq \frac{\epsilon^p \|a\|^p}{p} + \frac{\|b\|^q}{q\epsilon^q}, \tag{4}$$

*where $\epsilon > 0$, $p > 1$, and $q > 1$.*

**Lemma 2** ([41,42]). *For the filter with the following form:*

$$\dot{x}_O + \alpha(x_O - x_I) + \beta(x_O - x_I)^{q/p} = 0$$
$$x_O(0) = x_I(0), \tag{5}$$

*the output signal $x_O$ will track the input signal $x_I$ in a finite time, and the upper bound of time satisfies*

$$t = \frac{p}{\alpha(p-q)}(\ln[\alpha(x_O(0) - x_{I,\max})^{(p-q)/p} + \beta] - \ln\beta). \tag{6}$$

*where $\alpha$ and $\beta$ are parameters to be designed, and $p$ and $q$ are odd numbers and satisfy $p > q > 0$.*

**Lemma 3.** *The following inequality holds for the constant vector $b_1 > 0$ and any vector $x$ in the interval $|x| < |b|$:*

$$\log\frac{b^T b}{b^T b - x^T x} \leq \frac{x^T x}{b^T b - x^T x}. \tag{7}$$

**Assumption 1.** *The desired trajectory $y_d$ and its first derivative are available and bounded.*

**Assumption 2** ([43]). *The disturbance $\Lambda(t)$ is continuous and bounded, and its first order derivative satisfies $\Lambda(t) < \bar{\Lambda}$, $d(\Lambda(t))/dt < b_1$.*

**Assumption 3** ([44]). *For the actuator failure $\phi$, $\exists \bar{\phi} > 0$ such that $\phi < \bar{\phi}$ for $t > 0$.*

The dynamics model of an n-links robotic system with actuator failures can described as

$$M(q)\ddot{q} + C(q, \dot{q})\dot{q} + G(q) = \tau_{SD} + \Lambda(t) + \phi(q, \dot{q}, t) \tag{8}$$

where $q, \dot{q}, \ddot{q} \in R^n$ are the position, velocity, and acceleration vectors, respectively, $\tau_{SD} \in R^n$ is an input of the system, which is influenced by the saturation and dead zone, and $v \in R^n$ is an intermediate variable. $M(q) \in R^{n \times n}$ is an inertia matrix, $C(q, \dot{q}) \in R^{n \times n}$ denotes the Centripetal and Coriolis torques matrix, and $G(q) \in R^n$ represents the gravitational force vector. $\Lambda(t) \in R^n$ is a disturbance, and $\phi(q, \dot{q}, t)$ represents the fault function of the actuator during operation. Then, we use $M, C, G, \Lambda$, and $\phi$ to simplify the design below.

**Property 1.** *The matrix $M(q)$ is symmetric and positive definite.*

**Property 2.** *The matrix $\dot{M}(q) - 2C(q, \dot{q})$ is skew symmetric.*

Since actuator failures often occur in real systems, the description function is introduced and expressed as

$$\phi(t) = -D(t)\tau_{SD} + \phi_d, \tag{9}$$

where $\tau_{SD}$ is the input signal of the actuator, and $\phi_d$ means the uncertain deviation fault. $D(t) = \text{diag}(D_1, D_2 \ldots)$ represents the actuator effectiveness, with each element satisfying the condition $0 < D_i < 1$; when $D_i = 1$, it means a complete failure and $D_i = 0$ means no performance failure.

Let $x_1 = q$ and $x_2 = \dot{q}$, and then the description of the robot system can be rewritten as

$$\begin{aligned} \dot{x}_1 &= x_2 \\ \dot{x}_2 &= M^{-1}[\tau_{SD} + \Lambda + \phi - C\dot{x}_1 - G]. \end{aligned} \tag{10}$$

In practice, the actuator is often subjected to a variety of constraints, and in this paper, we consider the robot system with the dead zone and saturation constraints, along with actuator failures. Then, we employ a smooth function to describe the saturation and dead zone characteristics, which is designed as

$$\tau_{SD} = u_M \tanh\left(\frac{\iota(v)}{u_M}\right) = u_M \frac{e^{\iota(v)/u_M} - e^{-\iota(v)/u_M}}{e^{\iota(v)/u_M} + e^{-\iota(v)/u_M}}, \tag{11}$$

and

$$\iota(v) = kv + \frac{k}{2r} \ln\left(\frac{\cosh(r(v - \zeta_r))}{\cosh(r(v + \zeta_l))}\right) + \frac{k}{2}(\zeta_l - \zeta_r), \tag{12}$$

where $u_M$ is the bound of $\tau_{SD}$. $k, \zeta_l$, and $\zeta_r$ represent the scale factor and the dead zone left and right points, and $r$ is a positive constant. $v(t)$ is an intermediate variable, and then we state the auxiliary system as

$$\dot{v} = -cv + \omega, \tag{13}$$

with $c > 0$.

In this section, we describe the manipulator system with multiple constraints and actuator faults. The control objective is that the system can track the desired trajectory $y_d$. In order to reduce the impact of constraints and unknown faults on the system performance during the control process, we need a reasonable design of $\omega$.

## 3. Control Design

*Adaptive Neural Dynamic Surface Controller Design*

**Step 1:** First, the tracking error $e_1$ and the second error $e_2$ are defined as

$$e_1 = x_1 - y_d. \tag{14}$$

$$e_2 = x_2 - a_1. \tag{15}$$

where $a_1$ is a virtual control. The specific form is chosen as

$$a_1 = -K_1 e_1 + \sum_{i=1}^{n} \frac{e_{1i} \dot{b}_{1i}}{b_{1i}} + \dot{y}_d, \tag{16}$$

where $K_1 = \text{diag}(K_{11}, K_{12}, \ldots, K_{1n})$ is a positive matrix. In order to achieve the control objectives, i.e., $|e_1| < |b_1|$, with $e_1 = [e_{11}, e_{12}, \ldots, e_{1n}]^T$ and $b_1 = [b_{11}, b_{12}, \ldots, b_{1n}]^T$, we construct the first Lyapunov function as

$$V_1 = \frac{1}{2} \sum_{i=1}^{n} \log \frac{b_{1i}^2}{b_{1i}^2 - e_{1i}^2}. \tag{17}$$

Then the derivative of (17) yields

$$\dot{V}_1 = \sum_{i=1}^{n} \left( -\frac{K_{1i} e_{1i}^2}{b_{1i}^2 - e_{1i}^2} + \frac{e_{1i} e_{2i}}{b_{1i}^2 - e_{1i}^2} \right). \tag{18}$$

**Step 2:** The derivative of $e_2 = [e_{21}, e_{22}, \ldots, e_{2n}]^T$ is expressed as

$$\dot{e}_2 = M^{-1}[\tau_{SD} + \Lambda + \phi - C\dot{x}_1 - G] - \dot{a}_1. \tag{19}$$

We define the error $y_2$ between the output signal $a_{2O}$ and the input signal $a_{2I}$ of the first-order filter as

$$y_2 = a_{2O} - a_{2I}, \tag{20}$$

and a new error signal $e_3 = [e_{31}, e_{32}, \ldots, e_{3n}]^T$ is given as

$$e_3 = \tau_{SD} - a_{2O}. \tag{21}$$

Then $\dot{e}_2$ is rewritten as

$$\dot{e}_2 = M^{-1}[e_3 + y_2 + a_{2I} + \Lambda + \phi - C x_2 - G] - \dot{a}_1. \tag{22}$$

Therefore, we design the virtual control law $a_{2I}$ as

$$\begin{aligned} a_{2I} = -K_2 e_2 - \left[ \frac{e_{11}}{b_{11}^2 - e_{11}^2}, \frac{e_{12}}{b_{12}^2 - e_{12}^2}, \ldots, \frac{e_{1n}}{b_{1n}^2 - e_{1n}^2} \right]^T \\ + C a_1 + G + M \dot{a}_1 - \bar{\Lambda} \text{sgn}(e_2) - \phi, \end{aligned} \tag{23}$$

where $K_2$ is the positive matrix with $K_2 = \text{diag}(K_{21}, K_{22}, \ldots, K_{2n})$.

In the above equation, we should know the upper bound $\bar{\Lambda}$ in order to eliminate the effect of disturbance that is difficult to obtain in practice. To solve this problem, we utilize the adaptive scheme to approximate $\Lambda$, and we define $\hat{\Lambda}$ to be an estimated value, thus we have the approximate error

$$\tilde{\Lambda} = \Lambda - \hat{\Lambda}. \tag{24}$$

Taking its derivative, we obtain

$$\dot{\tilde{\Lambda}} = \dot{\Lambda} - \dot{\hat{\Lambda}}. \tag{25}$$

Then we design the adaptive law as

$$\dot{\hat{\Lambda}} = \Gamma_{di}(e_{2i} + \delta_{di}\hat{\Lambda}), \tag{26}$$

where $\Gamma_{di}$ and $\delta_{di}$ are positive constants. The NNs are utilized to approximate the unknown parameter part of the system, and the control law is modified as

$$\begin{aligned} a_{2I} = & -K_2 e_2 - \left[\frac{e_{11}}{b_{11}^2 - e_{11}^2}, \frac{e_{12}}{b_{12}^2 - e_{12}^2}, \ldots\right]^T \\ & - \hat{\Lambda} + \hat{W}_\tau^T S_\tau(Z) + \hat{W}^T S(Z), \end{aligned} \tag{27}$$

and we define the approximate error $(\tilde{\bullet}) = (\bullet)^* - (\hat{\bullet})$ so that

$$\begin{aligned} W^{*T} S(Z) &= \hat{W}^T S(Z) - \varepsilon \\ &= Ca_1 + G + M\dot{a}_1 - \varepsilon, \end{aligned} \tag{28}$$

$$\begin{aligned} W_\tau^{*T} S(Z_\tau) &= \hat{W}_\tau^T S(Z_\tau) - \varepsilon_\tau \\ &= \phi - \varepsilon_\tau, \end{aligned} \tag{29}$$

where $Z = [x_1^T, x_2^T, a_1^T, \dot{a}_1^T]^T$, $Z_\tau = [x_1^T, x_2^T, e_1^T, \tau_{SD}]^T$. $\hat{W}^T S(Z), \hat{W}_\tau^T S(Z_\tau)$ are the outputs of the network, and the updating law is designed as

$$\dot{\hat{W}}_i = -\Gamma_i[S_i(Z)e_{2,i} + \sigma_i \hat{W}_i], \tag{30}$$

$$\dot{\hat{W}}_{\tau i} = -\Gamma_{\tau i}[S_i(Z_\tau)e_{2,i} + \sigma_{\tau i} \hat{W}_{\tau i}], \tag{31}$$

where $\Gamma_i = \Gamma_i^T > 0$, $\Gamma_{\tau i} = \Gamma_{\tau i}^T > 0$, and $\sigma_i, \sigma_{\tau i} > 0$.

Consider the second Lyapunov function candidate as

$$V_2 = V_1 + \frac{1}{2} e_2^T M(x_1) e_2 + \frac{1}{2} \tilde{\Lambda}^T \Gamma^d \tilde{\Lambda} + \frac{1}{2} \sum_{i=1}^{n} \tilde{W}_i^T \Gamma_i^{-1} \tilde{W}_i + \frac{1}{2} \sum_{i=1}^{n} \tilde{W}_{\tau i}^T \Gamma_{\tau i}^{-1} \tilde{W}_{\tau i}. \tag{32}$$

Taking the derivative of (32) yields

$$\begin{aligned} \dot{V}_2 \leq & -\sum_{i=1}^{n} \frac{K_{1i} e_{1i}^2}{b_{1i}^2 - z_{1i}^2} - e_2^T(K_2 - I_{n \times n}) e_2 \\ & + e_2^T y_2 + e_2^T e_3 - \sum_{i=1}^{n} \frac{\sigma_i}{2} \|\tilde{W}_i\|^2 - \sum_{i=1}^{n} \frac{\sigma_{\tau i}}{2} \|\tilde{W}_{\tau i}\|^2 \\ & + \sum_{i=1}^{n} \frac{\sigma_i}{2} \|W_i^*\|^2 + \sum_{i=1}^{n} \frac{\sigma_{\tau i}}{2} \|W_{\tau i}^*\|^2 + \frac{1}{2} \|\bar{\varepsilon}\|^2 + \frac{1}{2} \|\varepsilon_\tau\|^2 \\ & - \frac{1}{2} \sum_{i=1}^{n} \left(\delta_{di} - \Gamma_{di}^{-1}\right) \tilde{\Lambda}_i^2 + \frac{1}{2} \sum_{i=1}^{n} \Gamma_{di}^{-1} b_{1i}^2 + \frac{1}{2} \sum_{i=1}^{n} \delta_{di} \bar{\Lambda}_i^2. \end{aligned} \tag{33}$$

Then we obtain

$$\begin{aligned} \dot{V}_2 \leq & -\sum_{i=1}^{n} \frac{K_{1i} e_{1i}^2}{b_{1i}^2 - z_{1i}^2} - e_2^T(K_2 - I_{n \times n}) e_2 \\ & + e_2^T y_2 + e_2^T e_3 - \sum_{i=1}^{n} \frac{\sigma_i}{2} \|\tilde{W}_i\|^2 - \sum_{i=1}^{n} \frac{\sigma_{\tau i}}{2} \|\tilde{W}_{\tau i}\|^2 \\ & - \frac{1}{2} \sum_{i=1}^{n} \left(\delta_{di} - \Gamma_{di}^{-1}\right) \tilde{\Lambda}_i^2 + C_2, \end{aligned} \tag{34}$$

where

$$C_2 = \sum_{i=1}^{n} \frac{\sigma_i}{2} \|W_i^*\|^2 + \sum_{i=1}^{n} \frac{\sigma_{\tau i}}{2} \|W_{\tau i}^*\|^2 + \frac{1}{2}\|\bar{\varepsilon}\|^2 + \frac{1}{2}\|\varepsilon_\tau\|^2 + \frac{1}{2}\sum_{i=1}^{n}\Gamma_{di}^{-1}b_{1i}^2 + \frac{1}{2}\sum_{i=1}^{n}\delta_{di}\bar{\Lambda}_i^2 \quad (35)$$

**Step 3:** Consider the final Lyapunov function $V_3$ as

$$V_3 = V_2 + \frac{1}{2}e_3^T e_3 + \frac{1}{2}y_2^T y_2. \quad (36)$$

In order to obtain the derivative of $a_{2I}$, we pass the virtual control signal $a_{2I}$ through the finite-time first order filter with positive constants $\alpha_2$ and $\beta_2$ as

$$\dot{a}_{2O} = -\alpha_2(a_{2O} - a_{2I}) - \beta_2(a_{2O} - a_{2I})^{q/p}, a_{2O}(0) = a_{2I}(0). \quad (37)$$

We utilize the Nussbaum function to convert $\omega$

$$\begin{aligned} N(\chi) &= \chi^2 \cos(\chi), \\ \dot{\chi} &= \gamma_\chi e_3 \bar{\omega}, \\ \omega &= N(\chi)\bar{\omega}, \end{aligned} \quad (38)$$

where $\gamma_\chi > 0$ is positive constant, and $\bar{\omega}$ is an auxiliary control signal vector.
$\bar{\omega}$ is constructed as

$$\bar{\omega} = -K_3 e_3 - e_2 + p_{SD}cv + \dot{a}_{2O}, \quad (39)$$

where $K_3 = \text{diag}(K_{31}, K_{32}, \ldots, K_{3n})$ is the positive matrix and $p_{SD} = \text{diag}(\frac{\partial \tau_{SD1}}{\partial v_1}, \frac{\partial \tau_{SD2}}{\partial v_2}, \ldots, \frac{\partial \tau_{SDn}}{\partial v_n})$.

Then we take the derivative of $V_3$

$$\begin{aligned} \dot{V}_3 \leq &-\sum_{i=1}^{n} \frac{K_{1i}e_{1i}^2}{b_{1i}^2 - z_{1i}^2} - e_2^T(K_2 - \frac{3}{2}I_{n\times n})e_2 - e_3^T K_3 e_3 \\ &- \sum_{i=1}^{n}\frac{\sigma_i}{2}\|\tilde{W}_i\|^2 - \sum_{i=1}^{n}\frac{\sigma_{\tau i}}{2}\|\tilde{W}_{\tau i}\|^2 - \frac{1}{2}\sum_{i=1}^{n}\left(\delta_{di} - \Gamma_{di}^{-1}\right)\tilde{\Lambda}_i^2 \\ &+ \sum_{i=1}^{n}\frac{\dot{\chi}_i}{\gamma_{\chi i}}(p_{g_i^v}N_i(\chi_i) - 1) + (1 - \alpha_2)y_2^T y_2 + C_3, \end{aligned} \quad (40)$$

where $C_3 = C_2 + \frac{1}{2}\eta_2^T \eta_2$, and $\eta_2$ is the nonnegative continuous function such that

$$|\dot{a}_{2I}| \leq \eta_2(z_1, e_2, \hat{W}_i, y_d, \dot{y}_d, \ddot{y}_d). \quad (41)$$

Finally, we obtain

$$\dot{V}_3 \leq -\rho V_3 + \sum_{i=1}^{n}\frac{\dot{\chi}_i}{\gamma_{\chi i}}(p_{g_i^v}N_i(\chi_i) - 1) + C_3, \quad (42)$$

where

$$\begin{aligned} \rho = \min[&\min(2K_{1i}), \min(\frac{\lambda_{\min}(2K_2 - 3I_{n\times n})}{\lambda_{\max}(M)}), \min(2K_{3i}), \\ &\min(\frac{\sigma_i}{\lambda_{\max}(\Gamma_i^{-1})}), \min(\frac{\sigma_{\tau i}}{\lambda_{\max}(\Gamma_{\tau i}^{-1})}), \min(\frac{\delta_{di} - \Gamma_{di}^{-1}}{\lambda_{\max}\left(\Gamma_d^{-1}\right)}), 2(1 - \alpha_2)]. \end{aligned} \quad (43)$$

Integrating (42) gives

$$V_3(t) - V_3(0) \leq -\rho \int_0^t V_3(\tau)d\tau + O, \quad (44)$$

where $O = \int \sum_{i=1}^{n} \frac{\dot{\chi}_i}{\gamma_{\chi i}} (p_{g_i^v} N_i(\chi_i) - 1) dt$. From [45], $\chi$ is bounded, so $O$ is bounded, and in turn we determine that there exists an upper bound $Q$ for $V_3$ such that

$$\frac{1}{2} \sum_{i=1}^{n} \log \frac{b_{1i}^2}{b_{1i}^2 - e_{1i}^2} \leq V_3 \leq Q,$$

$$\frac{1}{2} e_2^T M e_2 \leq V_3 \leq Q. \tag{45}$$

Then the error signals $e_1$ and $e_2$ are kept within the compact set $\Omega_{e_1}$ and $\Omega_{e_2}$

$$\Omega_{e_1} := \left\{ e_{1i} \in \mathbb{R}^n \mid \|e_{1i}\| \leq \sqrt{b_{1i}^2(1 - e^{-2Q})} \right\},$$

$$\Omega_{e_2} := \left\{ e_2 \in \mathbb{R}^n \mid \|e_2\| \leq \sqrt{\frac{2Q}{\lambda_{\min}(M)}} \right\}. \tag{46}$$

Similarly, we can determine that the errors $e_3$, $\tilde{\Lambda}$, $\tilde{W}$, and $\tilde{W}_\tau$ are bounded. Therefore, we can conclude that the tracking errors and the approximate errors of the system converge to zero under the conditions of the proposed control law and suitable parameters.

## 4. Simulations

### 4.1. Robotic System Establishment

In the simulation part, using the robotic arm model in [46], the structure is shown in Figure 2. The descriptions of inertia matrix $M(q)$, Centripetal and Coriolis torques matrix $C(q, \dot{q})$, and gravitational force vector $G(q)$ are provided as

$$M(q) = \begin{bmatrix} M_{11} & M_{12} \\ M_{21} & M_{22} \end{bmatrix}, \tag{47}$$

$$C(q, \dot{q}) = \begin{bmatrix} C_{11} & C_{12} \\ C_{21} & C_{22} \end{bmatrix}, \tag{48}$$

$$G(q) = \begin{bmatrix} G_{11} \\ G_{21} \end{bmatrix}, \tag{49}$$

and

$$\begin{aligned}
M_{11} &= m_1 l_{c1}^2 + m_2 \left( l_1^2 + l_{c2}^2 + 2l_1 l_{c2} \cos q_2 \right) + I_1 + I_2, \\
M_{12} &= m_2 \left( l_{c2}^2 + l_1 l_{c2} \cos q_2 \right) + I_2, \\
M_{21} &= m_2 \left( l_{c2}^2 + l_1 l_{c2} \cos q_2 \right) + I_2, \\
M_{22} &= m_2 l_{c2}^2 + I_2, \\
C_{11} &= -m_2 l_1 l_{c2} \dot{q}_2 \sin q_2, \\
C_{12} &= -m_2 l_1 l_{c2} (\dot{q}_1 + \dot{q}_2) \sin q_2, \\
C_{21} &= m_2 l_1 l_{c2} \dot{q}_1 \sin q_2, \\
C_{22} &= 0, \\
G_{11} &= (m_1 l_{c2} + m_2 l_1) g \cos q_1 + m_2 l_{c2} g \cos(q_1 + q_2), \\
G_{21} &= m_2 l_{c2} g \cos(q_1 + q_2).
\end{aligned} \tag{50}$$

The structural parameters of the robot are shown in Table 1, and the initial state is set as

$$q_1(0) = 0, q_2(0) = 1, \dot{q}_1(0) = 1, \dot{q}_2(0) = 0. \tag{51}$$

**Table 1.** Parameters of the robot.

| Parameter | Description | Value |
|:---:|:---:|:---:|
| $m_1$ | Mass of link 1 | 2.00 kg |
| $m_2$ | Mass of link 2 | 0.85 kg |
| $l_1$ | Length of link 1 | 0.35 m |
| $l_2$ | Length of link 2 | 0.31 m |
| $I_1$ | Moment of inertia of link 1 | $\frac{1}{4}m_1 l_1^2$ kgm$^2$ |
| $I_2$ | Moment of inertia of link 2 | $\frac{1}{4}m_2 l_2^2$ kgm$^2$ |

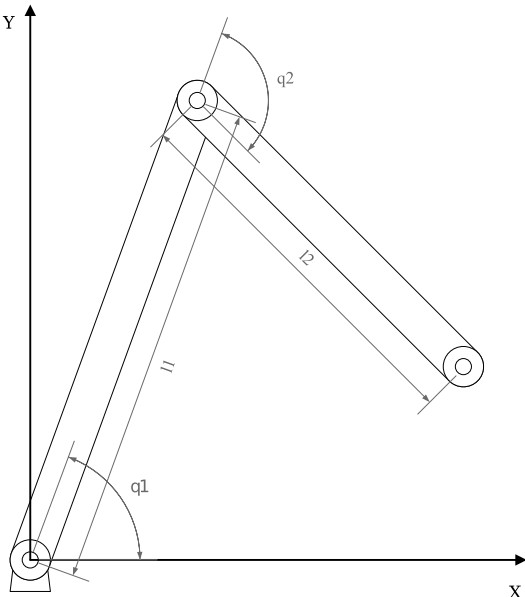

**Figure 2.** Sketch of two-link robot manipulator structure.

The desired tracking trajectory is given as $y_d = [\sin(t), \cos(t)]^T$, where $t \in [0, t_s]$ and $t_s = 20$ s. The other conditions and parameters are taken as $K_1 = \text{diag}(7,3)$, $K_2 = \text{diag}(10,5)$, $K_3 = \text{diag}(15,10)$, and $c = 2$. The disturbance is given as $\Lambda = [0.5\sin(t) + 1, 0.5\cos(t) + 0.5]^T$. Further, to describe the error signals during actuator operation, we consider the following form of $\phi$

$$\phi = \begin{cases} -0.4\tau_{SD}, & t \in [5,10] \\ -0.4\tau_{SD} + [2\cos(t), 2]^T. & t \in [13,18] \end{cases} \tag{52}$$

*4.2. Model-Based Control*

For the model-based (MB) control, we use the MB control designed in (23), and then the other parameters are set as $p = 15$, $q = 11$, $\alpha_2 = 40$, and $\beta_2 = 40$. The saturation value of the actuator $u_M$ is set as $\text{diag}(13,10)$. The dead zone characteristic parameters are set as $k = [1,1]^T$, $r = [2,3]^T$, $\zeta_l = [2,1]^T$, and $\zeta_r = [10,3]^T$, and the time-varying output constraints are $b_1 = [0.8\exp(-5t) + 0.2, 0.8\exp(-5t) + 0.2]^T$. The initial conditions are taken as $a_{2O}(0) = 0$, $\hat{\Lambda}(0) = [0,0]^T$.

The simulation results are illustrated in Figures 3–6. According to Figures 3 and 4, it can be seen that the system has good position tracking performance. The position output errors do not violate the time-varying output constraints, and the errors can be kept to a minimum during periods of actuator failure. Figures 5 and 6 illustrate the relationship between the system inputs and actuator inputs. The system input signals always stay within the saturation interval, and the control signals fluctuate widely at moments of sudden change.

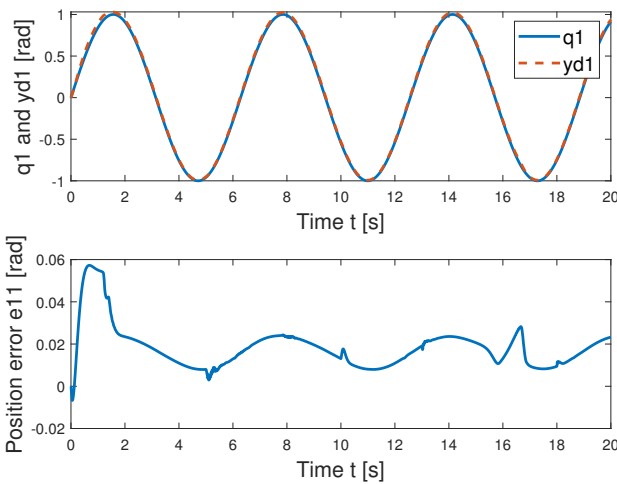

**Figure 3.** $q_1$ position trajectory and tracking error $e_{11}$ (MB control).

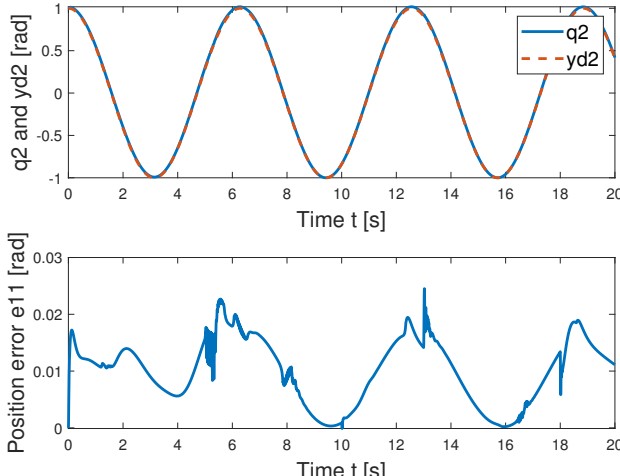

**Figure 4.** $q_2$ position trajectory and tracking error $e_{12}$ (MB control).

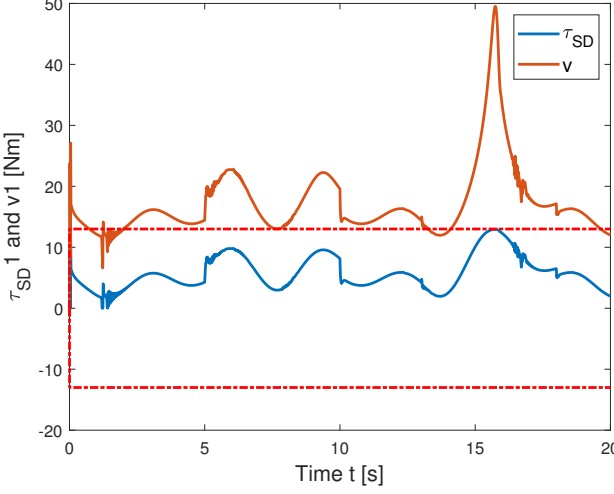

**Figure 5.** Control inputs $\tau_{SD}$ and $v$ for the first joint (MB control).

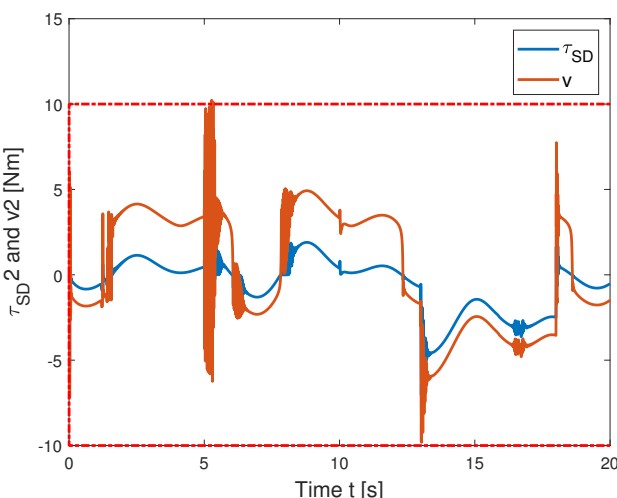

**Figure 6.** Control inputs $\tau_{SD}$ and $v$ for the second joint (MB control).

### 4.3. Adaptive Neural Network Control

For NNs control, we use the control law (27) with the update law (30) and (31). The constraint and gain parameters are the same as those of the MB control, and the control process is shown in Figure 7. The initial conditions are taken as $\hat{W}(0) = 0$, $\hat{W}_\tau(0) = 0$, $a_{2O}(0) = 0$, $\hat{\Lambda}(0) = [0,0]^T$. $S(Z)$ and $S(Z_\tau)$ both have 256 nodes; the node centers of NNs $\mu_{ki}, i = (1, 2, \ldots, 8)$ are selected in the area of $[-1, 1] \times [-1, 1] \times [-1, 1] \times [-1, 1] \times [-1, 1] \times [-1, 1] \times [-1, 1] \times [-1, 1]$, $\eta_k = 2$. The other parameters are given as

$$\Gamma = 10I_{256 \times 256}, \sigma = [0.01; 0.01],$$
$$\Gamma_\tau = 50I_{256 \times 256}, \sigma_\tau = [0.02; 0.02]. \tag{53}$$

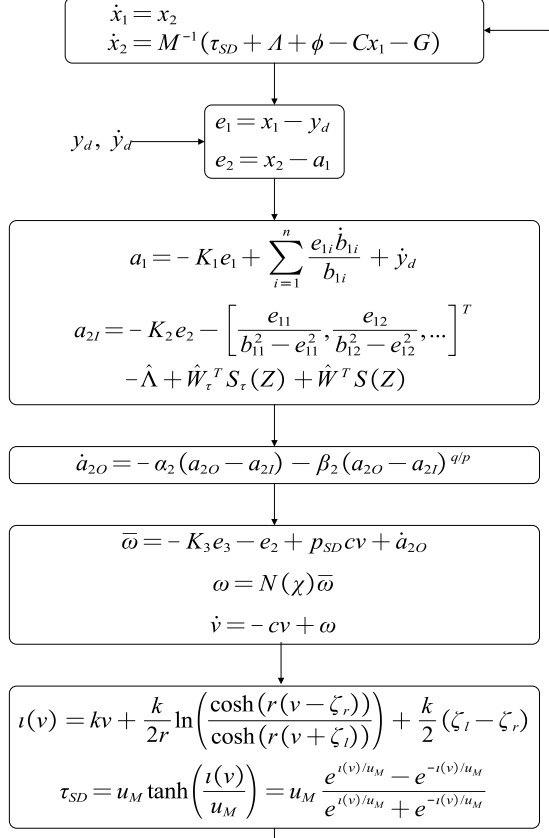

**Figure 7.** The control process of the adaptive neural network control.

The position tracking performance is shown in Figures 8 and 9, while the corresponding actuator input $v$ and the system input $\tau_{SD}$ are displayed in Figures 10 and 11. Compared with the control strategy of MB, the neural network-based control method reduces the fluctuation of the control signal to a certain extent. Figures 12 and 13 depict the weights of the NNs approximation $W$ and $W_\tau$.

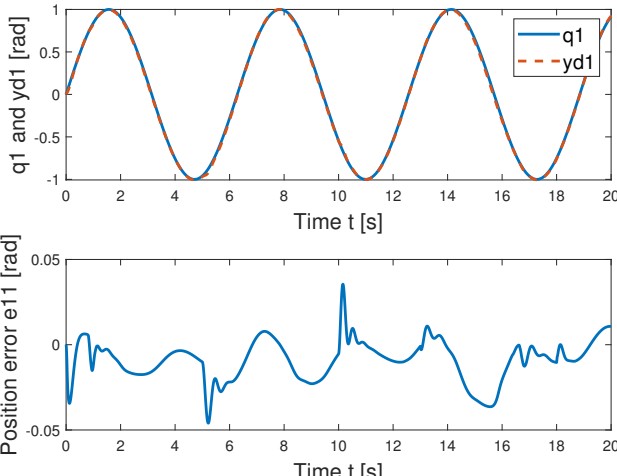

**Figure 8.** $q_1$ position trajectory and tracking error $e_{11}$ (NNs-based control).

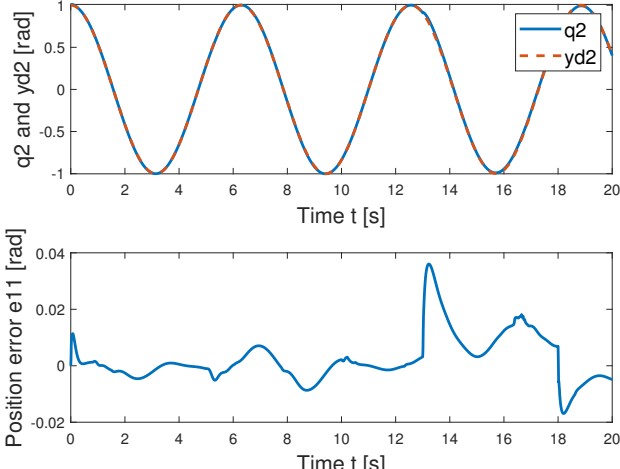

**Figure 9.** $q_2$ position trajectory and tracking error $e_{12}$ (NNs-based control).

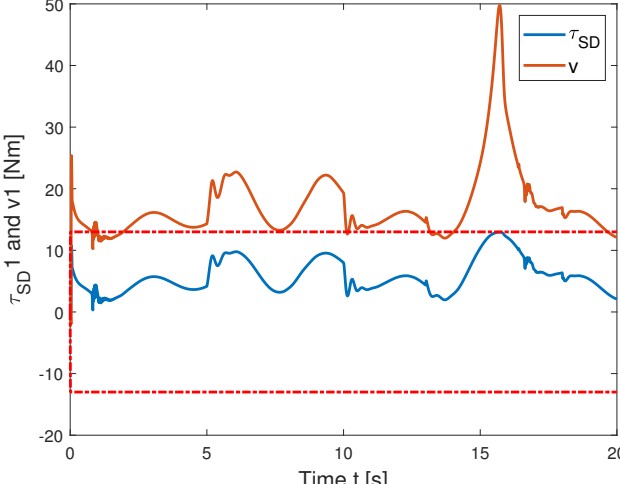

**Figure 10.** Control inputs $\tau_{SD}$ and $v$ for the first joint (NNs-based control).

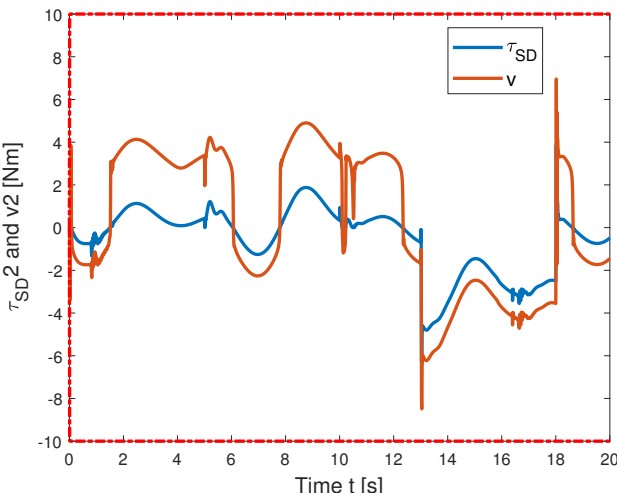

**Figure 11.** Control inputs $\tau_{SD}$ and $v$ for the second joint (NNs-based control).

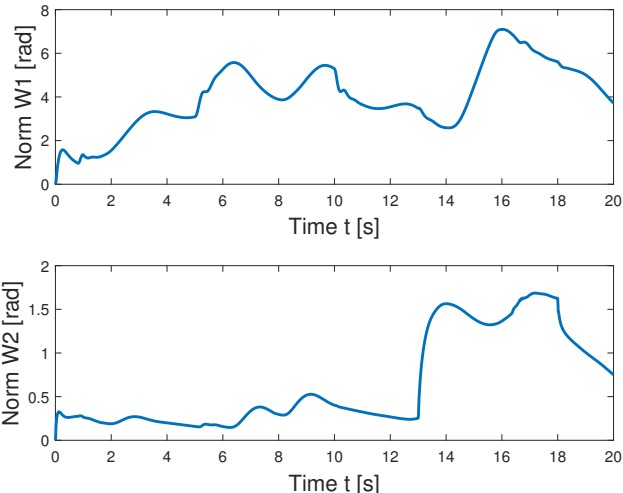

**Figure 12.** Norms of the adaptation weights $W$ (NNs-based control).

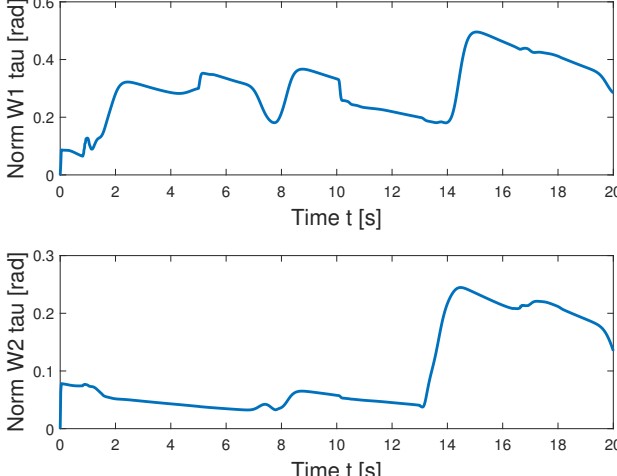

**Figure 13.** Norms of the adaptation weights $W_\tau$ (NNs-based control).

*4.4. PD Control*

This case considers a PD controller with a specific control law as $\tau = -K_p e_1 - K_d \dot{e}_1$, where $K_p = \mathrm{diag}(150, 20), K_d = \mathrm{diag}(50, 20)$. We compare the tracking errors for several

different cases with the same control constraints (uncompensated for faults, MB control (23), fault-tolerant DSC (F-DSC) ($\beta_2 = 0$), fault-tolerant finite-time DSC (F-FDSC) (27), and PD control) in Figures 14 and 15. First, when compared to the uncompensated, the employment of the NNs approach to compensate for the fault signal improves system performance during the fault time. Second, when the actuator's state changes, F-FDSC can make the tracking error enter the steady state more smoothly and reduce the error fluctuation compared to F-DSC.

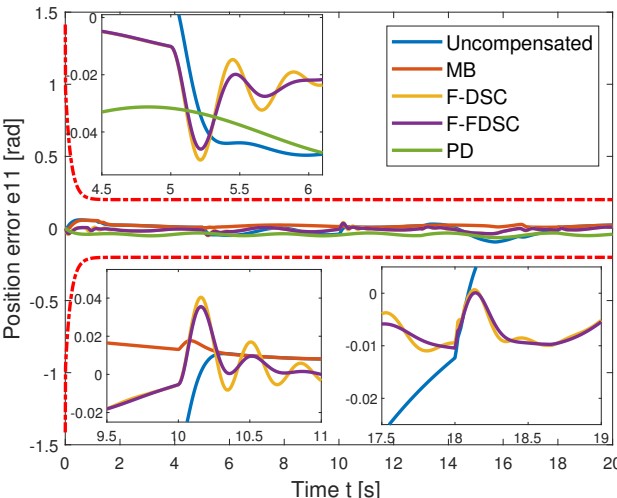

**Figure 14.** Position error $e_{11}$.

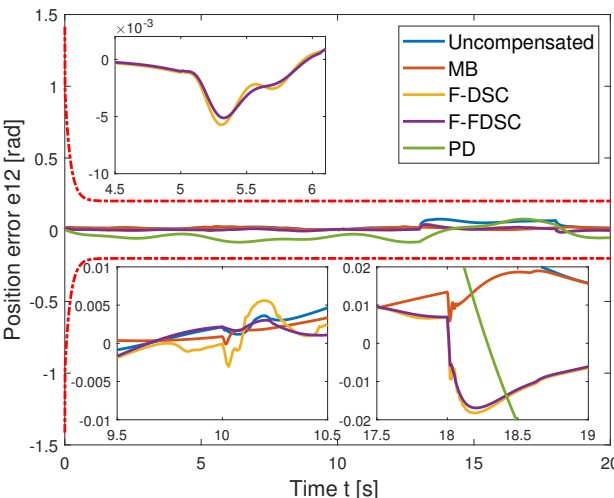

**Figure 15.** Position error $e_{12}$.

Finally, we sorted out in Table 2 the maximum tracking error values $E$ for the above methods during the period of actuator failure, where $E = \max|x_1 - y_d|$. By comparison, it can be seen that NNs fault-tolerant control is better than the traditional PD control.

**Table 2.** Output error values.

| Parameter | $e_{11}$ [rad] | $e_{12}$ [rad] |
|---|---|---|
| Uncompensated | 0.0936 | 0.0738 |
| MB | 0.0282 | 0.0245 |
| F-DSC | 0.0498 | 0.0366 |
| F-FDSC | 0.0460 | 0.0360 |
| PD | 0.0635 | 0.0872 |

## 5. Conclusions

In this study, a neural-network-based fault-tolerant controller was proposed for a robotic manipulator system with multiple constraints and actuator failures. A finite-time DSC filter was employed to optimize the design process and ensure that the output of the system converged quickly, even if actuator failure occurred. The neural network approach was applied to approximate actuator faults and uncertain robotic parameters, and a disturbance observer was used to eliminate the effects of external disturbances. Finally, the effectiveness of the proposed controller was demonstrated by the simulation results, that is, the system remained stable and the constraints were never violated, it had better performance during the actuator failure period, and the error signal could enter the steady state faster. The digital simulation initially verifies the feasibility of the designed controller, which we will also verify in real systems in the future.

**Author Contributions:** Conceptualization, Z.Z. and J.Z.; Data curation, Z.Z. and J.Z.; Funding acquisition, X.W.; Investigation, Z.Z.; Methodology, Z.Z.; Software, Z.Z.; Validation, L.P. and X.W.; Writing—original draft, Z.Z.; Writing—review and editing, L.P. and X.W. All authors have read and agreed to the published version of the manuscript.

**Funding:** This work was supported by the Scientific Research Projects of Guangzhou Education Bureau under Grant 202032793.

**Institutional Review Board Statement:** Not applicable.

**Informed Consent Statement:** Not applicable.

**Data Availability Statement:** Not applicable.

**Conflicts of Interest:** The authors declare no conflict of interest.

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
