# Peer review of "Finite-Time Neural Network Fault-Tolerant Control for Robotic Manipulators under Multiple Constraints"

_electronics, doi:10.3390/electronics11091343_

Round 1

Reviewer 1 Report

Dear authors,

Please correct the following issues.

1. Relation 7  states that x1' = x2' which the author already written they are x1=q and x2 = q'. Please review the equations and correct them where needed.

2. Some figures are too small for good visibility on the presented data. Please increase the size of figures 2-5 for better visibility of the overall error. Also, it looks that the position errors are quite small. For this, another diagram with zoom-in of figures 2, 3, 7, and 8 would be required for a better view of the error values.

3. Figures 4, 5, 9 and 10 present the speed tracking. Please add another diagram that would present the error zoomed in for a better view of the average tracking error, then explain the reason for the error spikes seen in figures 4, 5, 9, and 10.

4. Please add the unit of measurement in figures 2-11 and table 2. The visual information should be supported by precise data and notations.

5. Please add more arguments in the Conclusion section. In this section, you should also present the arguments to sustain your results and explain why the methods should be used on real robot applications. 

Thank you.

Author Response

Dear Reviewer,

Thank you very much for your valuable suggestions on our work. In response to your valuable comments, we have made changes and additions to the paper.

1.Comment: ``Relation 7  states that x1' = x2' which the author already written they are x1=q and x2 = q'. Please review the equations and correct them where needed."

       Response:~ We have made the check and change in Eq.10.

  1. Comment: ``Some figures are too small for good visibility on the presented data. Please increase the size of figures 2-5 for better visibility of the overall error. Also, it looks that the position errors are quite small. For this, another diagram with zoom-in of figures 2, 3, 7, and 8 would be required for a better view of the error values."

       Response:~ We have modified the structure of the figures and the size of the title, and we have changed the parameters of the simulation to make the results clearer.

  1. Comment: ``Figures 4, 5, 9 and 10 present the speed tracking. Please add another diagram that would present the error zoomed in for a better view of the average tracking error, then explain the reason for the error spikes seen in figures 4, 5, 9, and 10."

       Response:~First, the speed error fluctuation in the original figure is mainly caused by the failure of the actuator. Then, we have compared the results of some papers horizontally, and considering that our constraint is on the position output error, it is considered that the velocity tracking performance is not necessary to be discussed in this paper, so we have removed the related results and discussion.    

  1. Comment: ``Please add the unit of measurement in figures 2-11 and table 2. The visual information should be supported by precise data and notations."

       Response: ~We have made the changes in the figures 2~5, 7~14, and table 2.

  1. Comment: ``Please add more arguments in the Conclusion section. In this section, you should also present the arguments to sustain your results and explain why the methods should be used on real robot applications."

       Response: ~We have added several sets of experiments in the simulation section to further illustrate the effectiveness of the designed controller. And we have modified the content of the conclusion.

       Once again, thank you for your helpful comments.

Reviewer 2 Report

The reviewer has a few questions and comments:

  1. In (7), an error was apparently made in the first equation dx_1=dx_2
  2. It is not clear why the failure is described by function (6). What exactly is this representation about?
  3. In (8), (9) not all variables are defined.
  4. In section 2, it is not clear what is the problem statement after all? A description of the system is given, but what the problem itself is not indicated.
  5. The dimension of K1, e1, b1 in (13) is not indicated, it is “…”. What is y_d, a_2O?
  6. What is the requirement K_2 = K^T_2> 0? What is the reason for this requirement?
  7. The abbreviation NN has not been introduced after formula (23). There is no information about the architecture of the neural network, input and output data, learning algorithm, etc. The problem of approximating external disturbances using a neural network is quite complex in itself, but the authors do not provide information about its solution in the paper.
  8. In (25), (26) the notation is not clear (there is a comma, then an equal sign from a new line). Please clarify it.
  9. The initial conditions in (27), (28) are not set, how then is the differential equation solved?
  10. In figures 2-5, 7-10 variables x_11, x_d1, x_12, x_d2 are indicated. However, there is no explanation in the text what these variables are.
  11. Why is the value u_M = diag(50, 40) taken for the case of Adaptive Neural Network Control, what parameters were used for these constraint in other methods?
  12. Line 129 - instead of "Figs. 16 and 17", apparently it should be "Figs. 14 and 15".
  13. Section 4.4 discusses the comparison of three methods: PD, DSC, and finite-time DSC (FDSC). At the same time, line 132 refers to three methods: MB control, NN control and PD control. So what methods were still applied in the work? Such a presentation makes it difficult to accurately understand the results of the work, which is also due to the fact that Section 2 does not specify the problem to be solved. As a result, the presented description of the solution methods is also difficult to understand due to the fuzziness of the problem statement.

Author Response

Dear Reviewer,

Thank you very much for your valuable suggestions on our work. In response to your valuable comments, we have made changes and additions to the paper.

       1.Comment:``In (7), an error was apparently made in the first equation $dx_1=dx_2$."

       Response:~ We have made the check and change in Eq.10.

       2.Comment:``It is not clear why the failure is described by function (6). What exactly is this representation about?"

       Response:~ In the original we are missing a minus sign, and we have made a change in the corresponding section. This expression represents the lost efficiency during actuator operation, summed with the control signal $\tau_{SD}$, as the actual control input to the system.

       3.Comment:`` In (8), (9) not all variables are defined."

       Response:~    We have added the definition of $u_M$ after Eq.12.

       4.Comment:``In section 2, it is not clear what is the problem statement after all? A description of the system is given, but what the problem itself is not indicated."   

       Response:~    We have made changes, and at the end of Section 2, we summarize the issues involved and specify the control objectives.

       5.Comment:``The dimension of $K1, e1, b1$ in (13) is not indicated, it is $“…”$. What is $y_d, a_{2O}$?"

       Response:~    We have revised all dimension descriptions in the paper Eqs.(16),(23) and (39). and added the meanings of the letters in the equations, we have explained the meaning of $y_d$ and$ a_{2O}$ in line 100 and Eq.(20), respectively.

       6.Comment:``What is the requirement $K_2 = K^T_2> 0?$ What is the reason for this requirement?"

       Response:~    We have replaced the expression of $K_2$ as ``$K_2$ is the positive matrix with $K_2 = diag(K_{21}, K_{22}, ..., K_{2n})$''.

       7.Comment:``The abbreviation NN has not been introduced after formula (23). There is no information about the architecture of the neural network, input and output data, learning algorithm, etc. The problem of approximating external disturbances using a neural network is quite complex in itself, but the authors do not provide information about its solution in the paper."

       Response:~    We have added the abbreviations of the words in Section 1 (Line 30). And we have added an introduction to neural networks in Section 2, detailing information about the network structure, the input and output of the network, and the approximation error. And the results of several methods are compared in the simulation section to illustrate the rationality of neural networks.

       8.Comment:``In (25), (26) the notation is not clear (there is a comma, then an equal sign from a new line). Please clarify it."

       Response:~    We have removed unnecessary symbols from the equations.

       9.Comment:``The initial conditions in (27), (28) are not set, how then is the differential equation solved?"

       Response: We have added the values of some parameters and the settings of initial conditions in the simulation section.

       10.Comment:``In figures 2-5, 7-10 variables $x_{11}, x_{d1}, x_{12}, x_{d2}$ are indicated. However, there is no explanation in the text what these variables are."

       Response:~    We have made the changes in the figures 2, 3, 7 and 8. Corresponding symbols have been included for easy observation.

       11.Comment:``Why is the value $u_M = diag(50, 40)$ taken for the case of Adaptive Neural Network Control, what parameters were used for these constraint in other methods?"

       Response:~The magnitude of the saturation value will have an impact on the convergence of the errors. We intend to make the results more obvious, and we have modified the parameters in the simulation section to unify the constraint parameters and give further discussion in the simulation section.

       12.Comment:``Line 129 - instead of "Figs. 16 and 17", apparently it should be "Figs. 14 and 15"."

       Response:~    We have made changes to the presentation in the paper (Line 140).

       13.Comment:``Section 4.4 discusses the comparison of three methods: PD, DSC, and finite-time DSC (FDSC). At the same time, line 132 refers to three methods: MB control, NN control and PD control. So what methods were still applied in the work? Such a presentation makes it difficult to accurately understand the results of the work, which is also due to the fact that Section 2 does not specify the problem to be solved. As a result, the presented description of the solution methods is also difficult to understand due to the fuzziness of the problem statement."

       Response:~    We have redesigned the simulation part. Parameters have been modified and a control group has been added to better illustrate the problem, and we have described the control objectives and the issues considered in Section 2. Then the effectiveness of the controller has been further illustrated by simulation.

Once again, thank you for your helpful comments.

Reviewer 3 Report

The article deals with the actual scientific and practical task of developing methods for designing a control system for robotic systems, taking into account the limitations of actuators and possible failures.

Comments

1. The article does not consider the architecture of neural networks and the accuracy of approximation of nonlinear elements.
2. There are inaccuracies in the text (see line 62 and formula (7)).
3. From figure 1, the structure of the manipulator is not entirely clear.
4. The Simulation section does not present the structure of the control system and the architecture of the neural network.

Author Response

  1. Comment: ``The article does not consider the architecture of neural networks and the accuracy of approximation of nonlinear elements."

       Response:~ We have added an introduction to the principles and structure of neural networks in Section 2.

       2.Comment: ``There are inaccuracies in the text (see line 62 and formula (7))."

       Response:~ We have made the check and change in Eq.10, and the language issues in the paper were revised.

       3.Comment: ``From figure 1, the structure of the manipulator is not entirely clear."

       Response:~    We modified the structural sketch of the manipulator.

       4.Comment: ``The Simulation section does not present the structure of the control system and the architecture of the neural network."

       Response:~    We have added (Fig.6) to illustrate the control process and given details of our parameters in the simulation section.

Round 2

Reviewer 2 Report

In the revised version of the article, the authors took into account and corrected many technical comments. But, unfortunately, the reviewer still has a number of significant comments:

  1. The problem statement is still not clearly formulated. At the beginning of section 2, authors are talking about the problem of approximating some fault function using a neural network. In addition, at the end of the section, the control problem is reduced to choosing some variable w (omega). At the same time, there is not much connection between these statements.
  2. In response to Comment (vii): “… There is no information about the architecture of the neural network, input and output data, learning algorithm, etc. ...” the authors introduced a mathematical description of the problem of approximating some function f_i using a neural network. However, this does not answer the question posed about the architecture of a neural network, input and output data in relation to the considered problem, and not abstract mathematical relations explaining the principle of approximation using a neural network. It was assumed that the authors would give the architecture of a specific neural network used in their problem (specify the number of network layers; how many neurons to each layer; what were the inputs (what physical signals); if the fault function was applied to the input, then how in a real situation can it be distinguished from all external influences? What method was used to train the network? How much data was used in training, validation and testing?). In fact, there are no answers to these questions in the article.
  3. When modeling, several control options are considered: uncompensated for faults, MB control, fault-tolerant DSC (F-DSC), fault-tolerant finite-time DSC (F-FDSC), and PD control. At the same time, the mathematical basis of each control method is different. I It would be more convenient to analyze the results to clearly highlight the mathematical expressions for each of the considered control methods and, when comparing the results, consider the complexity of implementation. It may be that the accuracy of a simple PD controller is sufficient, while the implementation of the PD controller is simpler.

Technical remarks:

  1. In (1), (2) the index i=1,..,n is specified. In this case, the variable S has l elements, but in formulas (1), (2) there is an appeal to S_i. So what is S_i?

Author Response

Thank you for your valuable comments. We have provided a detailed response in the attached document.

Reviewer 3 Report

Replies have been received to all comments. 
Corrections have been made to the text of the paper.
I think that the paper can be published.

Author Response

Dear reviewer,

We would like to thank you again for your previous valuable comments on our work.

Round 3

Reviewer 2 Report

All the previously mentioned comments have been corrected by the authors